# SpCas9- and LbCas12a-Mediated DNA Editing Produce Different Gene Knockout Outcomes in Zebrafish Embryos

**DOI:** 10.3390/genes11070740

**Published:** 2020-07-03

**Authors:** Darya A. Meshalkina, Aleksei S. Glushchenko, Elana V. Kysil, Igor V. Mizgirev, Andrej Frolov

**Affiliations:** 1Sechenov Institute of Evolutional Physiology and Biochemistry, 194223 St. Petersburg, Russia; 2Department of Biochemistry, St. Petersburg State University, 199178 St. Petersburg, Russia; alexglushchenko11@gmail.com (A.S.G.); elana.kysil@gmail.com (E.V.K.); andrej.frolov@ipb-halle.de (A.F.); 3Department of Bioorganic Chemistry, Leibniz Institute of Plant Biochemistry, 06120 Halle (Saale), Germany; 4N.N. Petrov National Medical Research Center of Oncology, 197758 St. Petersburg, Russia; ivm1530@gmail.com

**Keywords:** CRISPR (Clustered Regularly Interspaced Short Palindromic Repeats), Cas9 (CRISPR associated protein 9), Cas12a (CRISPR associated protein 12a), Cpf1 (CRISPR-associated endonuclease in Prevotella and Francisella 1), zebrafish, gene knockout, repair outcome

## Abstract

CRISPR/Cas (Clustered Regularly Interspaced Short Palindromic Repeats/CRISPR associated protein) genome editing is a powerful technology widely used in current genetic research. In the most simple and straightforward way it can be applied for a gene knockout resulting from repair errors, induced by dsDNA cleavage by Cas nuclease. For decades, zebrafish (*Danio rerio*) has been known as a convenient model object of developmental biology. Both commonly used nucleases SpCas9 (*Streptococcus pyogenes* Cas9) and LbCas12a (*Lachnospiraceae bacterium* Cas12a) are extensively used in this model. Among them, LbCas12a is featured with higher specificity and efficiency of homology-directed editing in human cells and mouse. But the editing outcomes for these two nucleases in zebrafish are still not compared quantitatively. Therefore, to reveal possible advantages of one nuclease in comparison to the other in the context of gene knockout generation, we compare here the outcomes of repair of the DNA breaks introduced by these two commonly used nucleases in zebrafish embryos. To address this question, we microinjected the ribonucleoprotein complexes of the both nucleases with the corresponding guide RNAs in zebrafish zygotes and sequenced the target gene regions after three days of development. We found that LbCas12a editing resulted in longer deletions and more rare inserts, in comparison to those generated by SpCas9, while the editing efficiencies (percentage of mutated copies of the target gene to all gene copies in the embryo) of both nucleases were the same. On the other hand, overlapping of protospacers resulted in similarities in repair outcome, although they were cut by two different nucleases. Thus, our results indicate that the repair outcome depends both on the nuclease mode of action and on protospacer sequence.

## 1. Introduction

Being a simple and versatile tool for genetic studies, gene editing with CRISPR/Cas (Clustered Regularly Interspaced Short Palindromic Repeats/CRISPR associated protein) system has now became a research hotspot [1]. On the other hand, for decades, zebrafish (*Danio rerio*) has been known as a convenient model organism, especially useful in developmental genetics [2]. However, it is being rapidly established in other fields of science such as behavioral neuroscience, oncology, and pharmacology. With a range of advantages that zebrafish brings to these fields, the choice of adequate instrument for their genetic manipulation is a question important for obtaining more rapid results [3]. In this work, we have targeted zebrafish serotonin transporter genes (*slc6a4a* and *slc6a4b*), for which knockouts in rodents have proven their value in behavioral and pharmacological research [4,5]. Serotonin transporter limits the transmission of signal from the serotonergic cells and is the principle target of a broad class of antidepressants—selective serotonin reuptake inhibitors (for example, fluoxetin, citalopram, paroxetin, and sertraline). In contrast to mammals, zebrafish has two serotonin transporter genes, characterized with different localization of expression: *slc6a4a* is expressed in the raphe nuclei, ventral posterior tuberculum and pineal organ, whereas *slc6a4b* is expressed in the medulla oblongata and in the inner nuclear layer of retina [6]. This difference implies functional difference demanding further investigation by reverse genetics approaches.

CRISPR/Cas gene editing is based on RNA-guided recognition of the partly complementary sequence in DNA by nuclease and subsequent introduction of a break in the recognized site or nearby. *Streptococcus pyogenes* Cas9 (SpCas9) was one of the first nucleases used for CRISPR/Cas gene editing and it remains the most studied one, with a lot of engineered variants, optimized for various applications [7]. Cpf1 (or LbCas12a) from *Lachnospiraceae bacterium* is gaining its popularity due to its higher specificity and higher efficiency in homology-directed editing in human cells and mouse embryos [8,9,10]. These nucleases represent two different types of Cas proteins (II-a and V-a) and have different mechanisms of action [11]. SpCas9 forms complex with crRNA (crispr RNA) and tracrRNA (trans-activating crispr RNA) (that are sometimes fused into the single sgRNA molecule), starts its recognition with 5′-NGG-3′ PAM at the 3′-end of the protospacer region, and usually produces a blunt end, double-stranded DNA break very close to the protospacer adjacent motive (PAM, 3 bp upstream). However, it can also introduce single nucleotide overhangs by cutting 3 bp upstream PAM on the target strand and 4 bp upstream PAM on non-target strand [12]. LbCas12a forms complex with crRNA only, starts its recognition with 5′-TTTN-3′ PAM on the 5′-end of the protospacer, and cleaves the DNA strands asymmetrically and far from PAM (18 bp downstream on the non-target strand and 23 bp upstream on the target strand), producing sticky, single-stranded 5′-overhangs. This difference allows making supposition about larger deletion size after LbCas12a editing [13] (Figure 1).

This article aims to quantitatively confirm the supposition about the existence of characteristic nuclease-specific differences in deletion length, which requires a comparison of the properties of SpCas9 and LbCas12a in the context of generation of knockout zebrafish for behavioral research. Using Benchling [14] online service for gRNA site selection in the first exons of our target genes (*slc6a4a* and *slc6a4b*), we have introduced corresponding ribonucleoprotein (RNP) complexes into the zygotes of zebrafish before the first division. After embryo hatching, the gene regions adjacent to the target sites were sequenced. The obtained results of gene editing were characterized for each gRNA and each nuclease that allowed direct comparisons between them. We have found the efficiencies of both SpCas9 and LbCas12a in generating mutant alleles to be the same. However, the frequencies of insertions in the embryos edited with SpCas9 were higher, and the length of deletions introduced after LbCas12a double-stranded DNA break was increased.

## 2. Materials and Methods 

SpCas9 was purchased from NEB as EnGen^®^ Spy Cas9 NLS (nuclear localization signal, New England Biolabs, Ipswich, Massachusetts, USA). The procedure for extraction and purification of the NLS-fused LbCas12a was fully analogous to the protocol for CcCas9 described in Fedorova et al. [15], except the insertion of LbCas12a cDNA instead of CcCas9 cDNA into the pET21a plasmid. The guide RNAs were selected in the first exons of the target genes with the in-built Benchling service and listed in the Appendix A. crRNA and sgRNA were transcribed from the chemically synthesized and PCR-amplified templates using HiScribe™ T7 or Sp6 High Yield RNA Synthesis Kit (New England Biolabs, Ipswich, Massachusetts, USA) according to the manufacturer instructions. In vitro DNA cleavage assay was performed on the first exon PCR-products of *slc6a4a* and *slc6a4b* (the PCR conditions are described below), according to the protocol of Fedorova et al. [15], at 28 °C (the temperature of zebrafish embryo development) with the control incubated under the same reaction conditions without guide RNA.

Zebrafish strain AB was maintained in the ZebTEC automated housing system (Tecniplast, Buguggiate, Italy) in the Center of the Preclinical and Translational Research of Almazov Centre Institute of Experimental Medicine. In vitro fertilization was performed according to the protocol described in the Zebrafish Book [16] with modification of zebrafish sperm media from the cryopreservation protocol, which is described in details by Matthews et al. [17]. The obtained zygotes were microinjected with 2 nL of pre-formed RNP complexes with IM-300 microinjector (Narishige, Tokio, Japan) and micromanipulator MK-1 (Singer Instruments, Roadwater, UK) according to the protocol described by Rosen et al. [18]. The ribonucleoprotein (RNP) complexes were injected in the form of 1.5 μmol/L nuclease and 3 μmol/L solutions of corresponding crRNA or sgRNA in CutSmart Buffer (New England Biolabs, Ipswich, Massachusetts, USA). The embryos were raised to three days post-fertilization in E3 medium according to “The Zebrafish Book” [16]. We have found that controls of nuclease quality and embryo viability can be done by microinjections of RNP complexes of nuclease of interest with gRNA to *slc45a2* (melanocyte differentiation antigen) [19]: their protospacers had the sequences GACCGTACATACTCTTACTG for SpCas9 and GAAGGGAATTCTGCTACGCTGTT for LbCas12a. These control RNP complexes were microinjected to the separate group of embryos from the same clutch that was edited by the RNP complexes of interest. These embryos also served as specificity controls, as they were injected with the RNP complexes including gRNA, unrelated to the serotonin transporter genes.

Embryos at three days post-fertilization were euthanized by freezing and their DNA was extracted with Proteinase K (Helicon, Moscow, Russia) according to the protocol described in the Zebrafish Book [16] and purified on silica gel (Sigma, St. Louis, Missouri, USA). The numbers of the embryos from each group taken for the analysis are listed in the Appendix A. The fragments containing the first exons of the genes were PCR-amplified from 50 ng of genomic DNA with Tersus kit (Evrogen, Moscow, Russia). The primers for the *slc6a4a* first exon had the sequences F–GGACTGGTCACACTCTCCTTGC and R–CCCCCTCCATTTTGGTAGCAGATG. The primers for the *slc6a4b* first exon were used as F–AACTCTTGCTCAATCCTGAAGC and R–AAATTTCAGACGGCACTTTGAG (Figure 2a,b). The resulted PCR products were sequenced by fluorescent dye-terminator sequencing with Brilliant Dye kit (NimaGen, Nijmegen, Netherlands). The analysis relied on ABI 3500 Genetic Analyser (Thermo Fisher Sc., Waltham, Massachusetts, USA). The resulting sequences were analyzed with TIDE (Tracking of Indels by DEcomposition) online service [20], and for every target sequence the average efficiency, insertion and deletion ratios, indel ratio larger than 10 bp, and ratio of frameshift-free mutations were determined. Thereby, efficiency was considered as a ratio of sequence trace corresponding to mutated alleles of target genes to all sequence trace signal. Average deletion length was calculated for each embryo as a weighted average of all deletion lengths. Large deletion ratio was calculated as a percentage of a signal, obtained from Sanger sequencing, corresponding to deletions larger than 10 bp. Statistical comparison was performed with a Kruskal–Wallis test with post-hoc Dunn test and false discovery rate correction with R-Studio program package [21].

## 3. Results

### 3.1. crRNA and sgRNA Selection and In Vitro Efficiency Assessment

For selection the target sites in the first exons of genes *slc6a4a* and *slc6a4b* we relied on the commonly used Benchling service [14], focusing on the highest efficiency and specificity scores provided by the service [22,23]. The list of the target sites along with the scores is provided in Appendix A along with the scores. The locations of target sites in the first exons of *slc6a4a* and *slc6a4b* are shown in Figure 2a,b. For LbCas12a, an on-target score was not available, so we performed in vitro activity assessment for the selected and synthesized crRNA. We incubated preformed RNP complexes with the PCR products of the first exons of target genes and found that all of the selected crRNA variants were highly active and cut more than 90% of PCR product, except for slc6a4b-crRNA1 (Figure 2c). For this reason, all the analyzed crRNA were taken for the in vivo testing.

Besides, we synthesized marker crRNA and sgRNA to the first exon of melanocyte differentiation antigen *slc45a2* that was shown to be active in zebrafish by Moreno-Matteos et al. [19] and it could, therefore, provide a good visual control of nuclease ability to perform gene knockouts three days after RNP complexes were injected into the zygotes (Figure 2d). Knockout of this gene results in complete or mosaic loss of pigmentation, which is visible starting from the third day post-fertilization. We have included a group of embryos with such microinjections as controls in our subsequent experiments to verify proper nuclease and RNA storage conditions (with conservation of their activity) and viability of the embryos in the clutch sufficient to tolerate microinjections of RNP complexes. At the same time, these control embryos served as a specificity control, as the embryos were edited by the RNP complexes programmed with gRNA unrelated to the first exons of serotonin transporter genes.

### 3.2. Comparison of the ResultingMutants

Selected crRNA and sgRNA were proven to be effective in the *slc6a4a* and *slc6a4b* genes, except for slc6a4b-crRNA2, which did not demonstrate in vivo activity in the induction of mutations (Appendix A), though it had rather high in vitro activity (Figure 2a). Taking into consideration, that the control embryos injected in parallel with slc45a2-crRNA ribonucleoprotein complex with the same aliquot of LbCas12a demonstrated obvious phenotypic disturbance in the body coloration (Figure 2b), we can attribute this lack of activity to the individual property of this crRNA. Because of this reason, these embryos were not taken in the statistical analysis.

The target regions sequencing of 97 mutated embryos revealed their high mosaicism generated by SpCas9 and LbCas12a. On the other hand, an essential part of these embryos did not show any wild-type signal that implied they were mosaic in mutations. Thus, each embryo provided a spectrum of mutations for analysis (meanwhile, only indels less than 50 bp were available for analysis with TIDE service [20]). To characterize the restriction outcome for each type of RNP in particular and for each nuclease in general, we have averaged the data about the probability of each mutation type. 

We found that SpCas9 cuts result in more frequent insertions (*p* = 0.00017) in comparison to LbCas12a (Figure 3a). Thus, we confirmed the observation that LbCas12a produces longer deletions than SpCas9 (Figure 3b,c), increasing both the percentage of deletions larger than 10 bp (*p* < 0.00001) and average length of deletions (*p* < 0.00001). On the other hand, we found that slc6a4b-sgRNA4 in complex with SpCas9 produced deletions with the length not significantly different from those generated by LbCas12a (*p* = 0.46119). However, their length differed from the deletions produced by complexes of SpCas9 with other sgRNAs (all *p* < 0.05 with false discovery rate correction). The distinction of this sgRNA was not found for other outcome properties studied in our work that allows attributing this feature to peculiarities of the protospacer and surrounding sequences directing the repair to large deletions.

We found that efficiencies of both types of LbCas12a and SpCas9 RNP complexes in mutation introduction in wild-type alleles were almost the same (Figure 3d), while LbCas12a produced a higher outcome variance, presumably depending on the protospacer sequence.

## 4. Discussion

In this work, we analyzed the outcomes of SpCas9 and LbCas12a gene editing that can be used for further applications in knockout of zebrafish genes. This strategy is rather common in functional genomics research, so we selected the targets, requiring this approach – zebrafish serotonin transporter genes *slc6a4a* and *slc6a4b*, regulating serotonergic signaling [6]. Knockouts of these genes have a long history of usage for affective disorder modeling in rodents, valuable for the search of new antidepressants [4,5]. CRISPR/Cas technology provides a means to edit gene sequences rapidly, straightforwardly and cost-effectively in comparison to the previously available techniques. These properties, together with a high efficiency of editing [24] and versatility, provides an advantageous use of this method in research. However, it must be noted that projects dealing with therapeutic applications often rely on TALENs due to its high specificity and less dependence from the target site sequence [25].

We have applied TIDE analysis to decompose signal from Sanger sequencing chromatograms of PCR products and assess the spectra of indels from each embryo [20]. The decomposition is based on the alignment of two sequencing chromatograms of PCR products, amplified with the same primers from the wild-type and edited region of interest. The comparison of these sequence traces downstream the protospacer site allows to identify indels and their abundances in the edited PCR product. Low costs represent a serious advantage of the used approach in comparison to the technique of next-generation sequencing, currently considered as a gold standard. However, relative to the next-generation sequencing, TIDE was shown to underestimate the editing efficiencies and to have sensitivity limit of 2%, while being precise enough for the determination of indel size [26]. TIDE was previously used to estimate both the efficiencies [27,28] and features [29,30] of the editing outcome.

Several authors have previously reported that LbCas12a produces longer deletions in cell cultures and plants in comparison to SpCas9 [8,13,31]. This can be explained by the difference in the staggered-end vs mostly blunt-end nuclease cleavage mode. For non-homologous end joining, 5′-overhangs of a five-base length generated by LbCas12a are trimmed by Artemis [32] to the blunt ends (that are immediately generated by SpCas9), providing 5 bp longer deletions, that are corresponding to the deletion analysis results presented here (Figure 3b). Another explanation may be in the difference in location of the nuclease cleavage site: while SpCas9 cuts in the region overlapping the seed region, LbCas12a cuts far from the seed region and the resulting mutations may leave the seed region unaffected. This may lead to multiple rounds of LbCas12a-mediated recognition and subsequent cuts [11]. However, a lower incidence rate of insertions for LbCas12a was only recently mentioned in the literature for CRISPR/Cas mediated elimination of infectious human immunodeficiency virus from human cell cultures [33]. It was shown thereby that all inserts introduced by LbCas12a were accompanied by deletions (authors termed these class of outcomes “delins”). In this work authors used plasmid and lentiviral vectors, providing prolonged expression of CRISPR/Cas system components that differed from our approach, based on introduction of RNP complexes with limited lifetime. The longer time of action may increase the number of editing events, resulting in the absence of pure insertions in the edited cell population and longer deletions, in comparison to those obtained in our work. However, TIDE analysis of PCR-products is not able to distinguish delins from pure insertions or deletions [20]; that is why we can only declare the lower incidence of insertions, induced by LbCas12a. 

The demonstrated difference in the editing outcomes may be important for the zebrafish knockout line establishment in several ways. First, while searching for indels with TIDE it should be taken into account that it has a default indel length limit setting of 10 bp. This setting should be increased, especially when working with LbCas12a in order to analyze most of mutations, because most of the deletions appear to be longer than 10 bp (Figure 3b). Second, longer deletions, obtained with LbCas12a, provide more degrees of freedom in designing the genotyping assay for the knockout line maintenance. Simple, stable, and robust genotyping assay is important, because breeding and genotyping of the experimental animals is often done in non-molecular biology laboratories and thus should produce reliable results even in suboptimal conditions.

Our results also support an important role of protospacer sequence in the editing outcome, which was already shown for human cell cultures [34]. RNP complex of slc6a4b-sgRNA4 with SpCas9 produced longer deletions than complexes of SpCas9 with other sgRNAs, which were more similar with deletions, produced by LbCas12a. This can be partly explained by the sequence overlap of this sgRNA with slc6a4b-crRNA1 that also leads to large deletions, characteristic for LbCas12a. On the other hand, sequence overlap that was found between two other pairs of protospacers (slc6a4a-crRNA1 with slc6a4a-sgRNA4 and slc6a4b-crRNA3 with slc6a4b-sgRNA6 (Figure 2a,b), did not result in the similarities in repair outcome. Further research is needed to distinguish the contribution of protospacer sequence from that of the nuclease cut mode in the repair outcome.

Our results support the importance of preparation of several (at least two) gRNAs for animal gene knockout production. The availability and reliability of activity predictions by online services is rapidly growing, allowing in silico gRNA selection. Additionally, ribonucleoprotein complexes allow in vitro activity assessment prior to the embryo injections. But gRNA selected in these preliminary tests may fail to produce knockouts with any significant efficiency in vivo (as it was shown for slc6a4b-crRNA2 that demonstrated good cleavage activity in vitro, but zero editing efficiency in vivo). Our work underlines the importance of the control embryo injections with the ribonucleoprotein complex resulting in a visually obvious body phenotype. We used the *slc45a2* (melanocyte differentiation antigen) gene, the knockout of which impairs melanocyte-dependent body pigmentation [19]. This control can provide valuable information about the correctness of embryo obtaining and microinjection procedure, especially at the beginning of the facility functioning.

## Figures and Tables

**Figure 1 genes-11-00740-f001:**
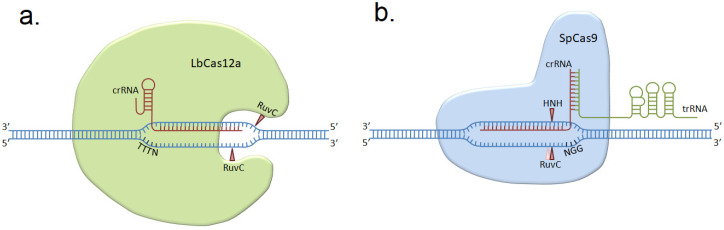
The mechanisms of action of LbCas12a (**a**) and SpCas9 (**b**) nucleases. LbCas12a cleaves far from PAM and SpCas9 cleaves near PAM, in the seed region. TTTN -represents the PAM sequence for LbCas12a, NGG is the PAM sequence for SpCas9, cleavage sites by nuclease domains (RuvC and HNH) are labeled with red triangles (pink triangle labels an alternative cut site, producing staggered ends).

**Figure 2 genes-11-00740-f002:**
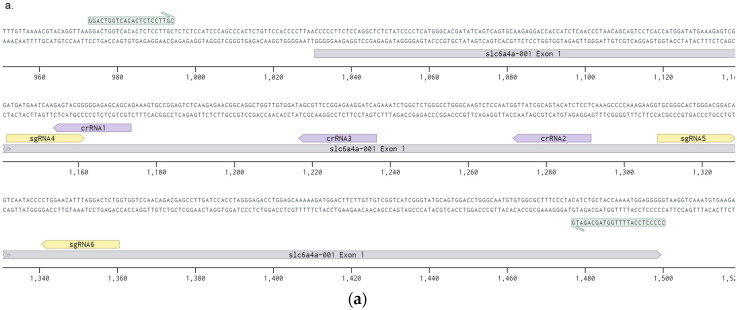
Assessment of crRNAs activities: (**a**) schematic representation of the protospacer location in the exon 1 of slc6a4a gene; (**b**) schematic representation of the protospacer location in the exon 1 of slc6a4b; (**c**) in vitro activity assessment of ribonucleoprotein complexes of LbCas12a with corresponding crRNAs: percentage of PCR-products that were cut by RNP complexes is given in the row under the picture; (**d**) representative phenotypes of RNP-microinjected embryos.

**Figure 3 genes-11-00740-f003:**
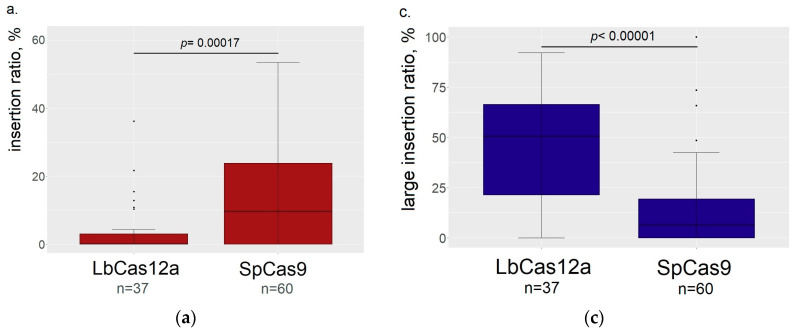
Comparison of LbCas12a and SpCas9 editing properties (the data from both serotonin transporters are summed together). (**a**) SpCas9 produces higher percentage of inserts in the outcomes of editing. (**b**) Average length of deletion is higher in embryos mutated by LbCas12a. (**c**) LbCas12a produces higher percentage of indels longer than 10 bp. (**d**) Both nucleases showed the same efficiency of editing.

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
