# Peer review of "SpCas9- and LbCas12a-Mediated DNA Editing Produce Different Gene Knockout Outcomes in Zebrafish Embryos"

_genes, 2020, doi:10.3390/genes11070740_

Round 1
Reviewer 1 Report
Major:
- Introduction: the authors did not describe and compare both editing tools/systems; there are many more important differences between Cas9 and Cas12a in terms of PAM, gRNA, DNA repair mechanisms involved in repair of different ends (blunt vs sticky).
- Introduction: line 32, SpCas9 was one of the first nucleases used for genome editing, the sentence is not precise (meganucleases, ZFN, TALENs, Streptococcus thermophilus Cas9 published by V. Siksnys group 2012),
- Introduction, lines 36-37: It has been reported that activity of SpCas9 protein results in both blunt and staggered ends. HNH and RuvC cleavage domains have ability to cut between 3|4 nt and 4|5 nt upstream of the PAM target and non- target DNA strands, respectively. Thus single nucleotide overhangs can be created.
- Introduction, lines 33-35: please add reference
- Materials and methods: detailed information about LbCas12a protein and protocol modifications should be presented,
- Experimental procedures, conditions, sequences of primers, etc. should be described in the materials and methods section,
- What was the control in in vitro cleavage assay?
- Results: line 88 – Figure 1b
- Figure 1b- slc45a2??
- The authors claim that: “Selected crRNA and sgRNA were proven to be effective in the slc6a4a and slc6a4b genes, except for slc6a4b-crRNA2 that did not demonstrate in vivo activity in the induction of mutations, though it had rather high in vitro activity”. Unfortunately we do not see any results (a table with editing efficiency data, western blot analysis of functional knockout, etc.…)
- Lines 105-106: again, please show some data
- Lines 121-122: “LbCas12a produced more outcome variance, presumably depending on the protospacer sequence” – a figure with marked target sites for both endonucleases/gRNAs would be helpful.
- Discussion: the difference between observed editing outcomes results probably from DNA repair mechanisms (c-NHEJ vs homology-based repair pathways). Recent study demonstrates that Cas12a does not induce DNA insertions that are routinely observed for Cas9 (Gao Z et al., 2020 Nucleic Acids Research, DOI: 10.1093/nar/gkaa226), but mainly deletions and deletions combined with small insertions. Please discuss your results in light of the new data concerning DNA repair and prediction of editing outcomes.
Author Response
Introduction: the authors did not describe and compare both editing tools/systems; there are many more important differences between Cas9 and Cas12a in terms of PAM, gRNA, DNA repair mechanisms involved in repair of different ends (blunt vs sticky).
Authors thank the Reviewer for this valuable comment. In our introduction we tried to keep it as concise as possible. But as the expansion of the introduction can clearly provide more profound background to our work, we have done it, following accurately to the reviewer comments.
“SpCas9 forms complex with crRNA and tracrRNA (that are sometimes fused into the single sgRNA molecule), starts its recognition with 5’-NGG-3’ PAM on the 3’-end of the protospacer region, and usually produces blunt-end double-stranded DNA break very close to PAM (3 bp upstream). However, it can also introduce single nucleotide overhangs by cutting 3 bp upstream PAM on the target strand and 4 bp upstream PAM on non-target strand [12]. LbCas12a forms complex with crRNA only, starts its recognition with 5’-TTTN-3’ PAM on 5’-end of protospacer, and cleaves the DNA strands asymmetrically and far from PAM (18 bp downstream on the non-target strand and 23 bp upstream on the target strand), producing sticky single-stranded 5’-overhangs.” Lines 53-61
Introduction: line 32, SpCas9 was one of the first nucleases used for genome editing, the sentence is not precise (meganucleases, ZFN, TALENs, Streptococcus thermophilus Cas9 published by V. Siksnys group 2012),
We thank the Reviewer and had refined the statement according to the comment.
“Streptococcus pyogenes Cas9 (SpCas9) was one of the first nuclease used for CRISPR/Cas gene editing and it remains the most studied one, with a dozen of modifications available for all demands [7].” Lines 48-49
Introduction, lines 36-37: It has been reported that activity of SpCas9 protein results in both blunt and staggered ends. HNH and RuvC cleavage domains have ability to cut between 3|4 nt and 4|5 nt upstream of the PAM target and non- target DNA strands, respectively. Thus single nucleotide overhangs can be created.
We thank the Reviewer for this valuable comment and have discussed this point in the introduction.
“However, it can also introduce single nucleotide overhangs by cutting 3 bp upstream PAM on the target strand and 4 bp upstream PAM on non-target strand [12].” Lines 56-57
Introduction, lines 33-35: please add reference
We have added the reference for higher specificity of LbCpf1 over SpCas12a.
“Cpf1 (or LbCas12a) from Lachnospiraceae bacterium is gaining its popularity due to its higher specificity and higher efficiency in homology directed editing in human cells and mouse embryos [8-10].” Lines 50-52
Materials and methods: detailed information about LbCas12a protein and protocol modifications should be presented,
We have added a clarification in the text of the article, that the only difference of our protocol from the reference purification protocol was in plasmid construction, containing LbCas12a gene insert instead of CcCas9.
“LbCas12a fused with NLS extraction and purification was fully analogous to the protocol for CcCas9 described in Fedorova et al. [14] except the insertion of LbCas12a cDNA instead of CcCas9 cDNA into the pET21a plasmid.” Lines 75-76
Experimental procedures, conditions, sequences of primers, etc. should be described in the materials and methods section,
We have thoroughly revised the Methods section, including the requested details.
“We have found that controls of nuclease quality and embryo viability can be done by microinjections of RNP complexes of nuclease of interest with gRNAs to slc45a2 (melanocyte differentiation antigen) [18]: their protospacers are GACCGTACATACTCTTACTG for SpCas9 and GAAGGGAATTCTGCTACGCTGTT for LbCas12a.” lines 93-97
“Primers for slc6a4a first exone were: F – GGACTGGTCACACTCTCCTTGC, R – CCCCCTCCATTTTGGTAGCAGATG; primers for slc6a4b first exone were: F – AACTCTTGCTCAATCCTGAAGC, R – AAATTTCAGACGGCACTTTGAG (Figure 1a and 1b).” lines 105-108
What was the control in in vitro cleavage assay?
We have included the description of the control reaction in the description of the in vitro cleavage assay.
“In vitro DNA cleavage assay was performed on the first exone PCR-products of slc6a4a and slc6a4b (PCR conditions are described below) according to protocol in Fedorova et al. [14] at 28°C (the temperature of zebrafish embryo development) with control incubated in the same reaction conditions without guide RNA.” Line 83
Results: line 88 – Figure 1b Figure 1b- slc45a2??
The text of the Results describes microinjections of RNP complex of nuclease of interest and crRNA for slc45a2 in the zygotes of the same clutch alongside with the microinjections of the RNP complexes of interest. We found this control to be very important for two reasons: 1. Control of the embryo viability in the clutch. 2. Control of the nuclease quality – if there is no albino or albino mosaic embryo among the microinjected with the RNP complex, than we can conclude nuclease have lost its activity (that may happen, for example, because of the compromised storage conditions). We have expanded the explanation of this control in the manuscript text.
“We have found that controls of nuclease quality and embryo viability can be done by microinjections of RNP complexes of nuclease of interest with gRNAs to slc45a2 (melanocyte differentiation antigen) [18]: their protospacers are GACCGTACATACTCTTACTG for SpCas9 and GAAGGGAATTCTGCTACGCTGTT for LbCas12a. Microinjections of these control RNP complexes were done to the separate group of embryos from the same clutch that was edited by the RNP complexes of interest. These embyos also served as specificity controls, as they were injected with the RNP complexes including gRNA, unrelated to the serotonin transporter genes.” Lines 93-100
The authors claim that: “Selected crRNA and sgRNA were proven to be effective in the slc6a4a and slc6a4b genes, except for slc6a4b-crRNA2 that did not demonstrate in vivo activity in the induction of mutations, though it had rather high in vitro activity”. Unfortunately we do not see any results (a table with editing efficiency data, western blot analysis of functional knockout, etc.…)
We have included the boxplot with the editing efficiencies data for all groups of embryos in the Supplement (Figure S1) and we have expanded the explanation of the fact slc6a4b-crRNA2 group was not taken into analysis.
“Selected crRNA and sgRNA were proven to be effective in the slc6a4a and slc6a4b genes, except for slc6a4b-crRNA2 that did not demonstrate in vivo activity in the induction of mutations (Figure S1), though it had rather high in vitro activity (Figure 1a). Taking into consideration that the control embryos injected in parallel with slc45a2-crRNA ribonucleoprotein complex with the same aliquot of LbCas12a demonstrated obvious phenotypic disturbance in the body coloration (Figure 1d), we can attribute this lack of activity to the individual property of this crRNA. That was the reason why we did not take these embryos in the statistical analysis.” Lines 148-154
Lines 105-106: again, please show some data
We have expanded Figure 1d, and included the representative pictures of all groups of embryos.
Lines 121-122: “LbCas12a produced more outcome variance, presumably depending on the protospacer sequence” – a figure with marked target sites for both endonucleases/gRNAs would be helpful.
We have included figures 1a and 1b, illustrating target sites in both analyzed genes.
Discussion: the difference between observed editing outcomes results probably from DNA repair mechanisms (c-NHEJ vs homology-based repair pathways). Recent study demonstrates that Cas12a does not induce DNA insertions that are routinely observed for Cas9 (Gao Z et al., 2020 Nucleic Acids Research, DOI: 10.1093/nar/gkaa226), but mainly deletions and deletions combined with small insertions. Please discuss your results in light of the new data concerning DNA repair and prediction of editing outcomes.
Authors thank the Reviewer for this valuable point, putting our results in the light of the new data. The article stated that all the insertions found in Cas12a edited samples were accompanied by the deletions. In the text of discussion we have compared the obtained results that made this section more comprehensive.
“But lower incidence of insertions for LbCas12a was only recently mentioned in the literature for human cell cultures [32], that has shown that all the inserts introduced by LbCas12a are accompanied by deletions. However, TIDE analysis of PCR-products is not able to distinguish delins from pure insertions or deletions [19] that is why we can only declare the lower incidence of insertions, induced by LbCas12a.” lines 212-217
Reviewer 2 Report
The present study describes a comparative analysis of two CRISPR/Cas gene editing approaches in zebrafish embryos. The authors provided necessary background information on the Cas systems utilized in this study, but other key information was neglected. The concerns below need to be addressed in order to improve the impact of the manuscript.
Major concerns:
- There is a common concern of specificity and off-target sites with the use of Cas nucleases that was not discussed in this manuscript. Please edit the manuscript to include these details.
- When publishing CRISPR/Cas data, authors should disclose the use of proper controls. It was not clear what controls were used in this study. Please revise and show include data using scrambled/random sgRNAs and/or SpCas9 and LbCas12a alone.
- The overall organization of the manuscript is lacking clarity to the reader. The target genes utilized in this study were not clearly defined until lines 138 in the discussion. Similarly, the control gene was not fully described until lines 171-174 in the discussion. These need to be moved to the introduction and section 3.2, respectively.
- Authors mention this study has a high impact in the field of gene editing and is especially advantageous in the zebrafish field, yet the authors never discuss the traditional approaches to incorporate gene KO in this model. Please edit the manuscript to include a literature review of the advantages of these gene editing approaches over TALENs, zinc finger nuclease, etc.
- What crRNA was used in Fig1b? It is not clear looking from Fig 1a to 1b which crRNA the authors moved forward with in vivo. Also, please include images of your sgRNA and/or SpCas9 and LbCas12a controls in vivo.
Minor concerns:
- Provide more background on target genes in introduction.
- Figure panels are not easily identifiable. Please edit figures to have panels A, B, etc. more visible (i.e. larger text, left placement).
- Lines 98-104 describe the in vivo efficacy of slc6a4b-crRNA2 compared to slc25a2-crRNA control. Where is this data represented? Please include an additional figure for this data or mention that the data is not shown.
Author Response
The present study describes a comparative analysis of two CRISPR/Cas gene editing approaches in zebrafish embryos. The authors provided necessary background information on the Cas systems utilized in this study, but other key information was neglected. The concerns below need to be addressed in order to improve the impact of the manuscript.
The authors thank the Reviewer for valuable comments and have corrected the manuscript accordingly.
When publishing CRISPR/Cas data, authors should disclose the use of proper controls. It was not clear what controls were used in this study. Please revise and show include data using scrambled/random sgRNAs and/or SpCas9 and LbCas12a alone.
We have added the detailed description of the control reactions for in vitro editing assay. We have expanded the description of control embryos microinjections with RNP complexes, including the nuclease of interest and gRNA targeting the melanocyte-differentiation antigen slc45a2, unrelated to the serotonin transporters genes. This gene knockout produced obvious coloration loss of embryos.
“In vitro DNA cleavage assay was performed on the first exone PCR-products of slc6a4a and slc6a4b (PCR conditions are described below) according to protocol in Fedorova et al. [14] at 28°C (the temperature of zebrafish embryo development) with control incubated in the same reaction conditions without guide RNA.” Line 83
“We have found that controls of nuclease quality and embryo viability can be done by microinjections of RNP complexes of nuclease of interest with gRNAs to slc45a2 (melanocyte differentiation antigen) [18]: their protospacers are GACCGTACATACTCTTACTG for SpCas9 and GAAGGGAATTCTGCTACGCTGTT for LbCas12a. Microinjections of these control RNP complexes were done to the separate group of embryos from the same clutch that was edited by the RNP complexes of interest. These embyos also served as specificity controls, as they were injected with the RNP complexes including gRNA, unrelated to the serotonin transporter genes.” Lines 93-100
The overall organization of the manuscript is lacking clarity to the reader. The target genes utilized in this study were not clearly defined until lines 138 in the discussion. Similarly, the control gene was not fully described until lines 171-174 in the discussion. These need to be moved to the introduction and section 3.2, respectively.
We thank the Reviewer for this comment, that improves the readability of the manuscript. We have clarified the text of our manuscript as required by the Reviewer: we defined the target genes in the Introduction and the control gene in the Methods sections.
“In this work we have targeted zebrafish serotonin transporters genes (slc6a4a and slc6a4b), which knockouts in rodents have proven their value in behavioral and pharmacological research [4, 5]. Serotonin transporter limits the transmission of signal from the serotonergic neurons and is the principle target of widely-spread class of antidepressants – selective serotonin reuptake inhibitors (for example, fluoxetin, citalopram, paroxetin, and sertraline). In contrast to mammals, zebrafish have two serotonin transporter genes, which were shown to have different localization of expression [6] that implies functional difference demanding further investigation by reverse genetics approaches.” Lines 38-45
Authors mention this study has a high impact in the field of gene editing and is especially advantageous in the zebrafish field, yet the authors never discuss the traditional approaches to incorporate gene KO in this model. Please edit the manuscript to include a literature review of the advantages of these gene editing approaches over TALENs, zinc finger nuclease, etc.
Authors thank the Reviewer for valuable point. We have added a reviewing paragraph to the Discussion.
“CRISPR/Cas technology provides means to edit gene sequences rapidly, straightforwardly and cost-effectively in comparison to the previously available instruments. These properties together with high efficiency of editing [23] and versatility provides an advantageous use of this method in research. However, it must be noted that projects of therapeutic applications often rely on TALENs due to its high specificity and less dependence from the target site sequence [24].” Lines 191-196
What crRNA was used in Fig1b? It is not clear looking from Fig 1a to 1b which crRNA the authors moved forward with in vivo. Also, please include images of your sgRNA and/or SpCas9 and LbCas12a controls in vivo.
We have clarified that we used all analyzed crRNA RNP complexes in subsequent microinjections in vivo in the text of the results, as all of them showed high efficiency in vitro cleavage assay.
“For that reason we have taken all of the analysed crRNA for the in vivo testing.” Line 127
Provide more background on target genes in introduction.
We have expanded the Introduction, describing the target genes and their importance for the investigation of the anxiety-related behavior in zebrafish.
“In this work we have targeted zebrafish serotonin transporters genes (slc6a4a and slc6a4b), which knockouts in rodents have proven their value in behavioral and pharmacological research [4, 5]. Serotonin transporter limits the transmission of signal from the serotonergic neurons and is the principle target of widely-spread class of antidepressants – selective serotonin reuptake inhibitors (for example, fluoxetin, citalopram, paroxetin, and sertraline). In contrast to mammals, zebrafish have two serotonin transporter genes, which were shown to have different localization of expression [6] that implies functional difference demanding further investigation by reverse genetics approaches.” Lines 38-45
Figure panels are not easily identifiable. Please edit figures to have panels A, B, etc. more visible (i.e. larger text, left placement).
In formatting figures for submission we followed the template provided by the Journal. However, we have tried to make the panels designations more visible.
Lines 98-104 describe the in vivo efficacy of slc6a4b-crRNA2 compared to slc25a2-crRNA control. Where is this data represented? Please include an additional figure for this data or mention that the data is not shown.
We have made the description of data selection for analysis more detailed and provided the group efficiency plot (Figure S1) to emphasize the efficiency of slc6a4b-crRNA2 on the first exone of slc6a4b was not found, that differed this gRNA from all other gRNA in the study (line 242).
Reviewer 3 Report
[Genes] Manuscript ID: genes-818996
Outcomes comparison of SpCas9- and LbCas12a-mediated DNA editing in zebrafish embryos
RECOMMENDATION - MINOR REVISION
MINOR COMMENTS
- Please provide detailed description of the Section ‘Materials and Methods”. The Authors should describe this section so that can be repeated by other scientists.
- The Authors should correct the sentence (lines 52-53) – “LbCas12a was extracted and purified accordingly to the previously published protocol with modifications”. What kind of modifications the Authors did?
- The Authors should correct the sentence (lines 59-60) – “In vitro fertilization was performed according to the protocol described in the Zebrafish Book with modifications for sperm media from the cryopreservation protocol” What kind of modifications the Authors did?
- Please provide the “N” number of animals / group sizes.
- There is no information about number of replicates.
- There is at least another paper on Cas12a editing in Zebrafish. The authors should discuss the other papers in the field with their results (Liu P, Luk K, Shin M, et al. Enhanced Cas12a editing in mammalian cells and zebrafish. Nucleic Acids Res. 2019;47(8):4169‐ doi:10.1093/nar/gkz184).
- Introduction and Discussion section are to short.
- The introduction section should end with the aim of the paper.

Author Response
Please provide detailed description of the Section ‘Materials and Methods”. The Authors should describe this section so that can be repeated by other scientists.
We have specified the differences we have made in the reference protocols.
The Authors should correct the sentence (lines 52-53) – “LbCas12a was extracted and purified accordingly to the previously published protocol with modifications”. What kind of modifications the Authors did?
We have described the only difference from the reference protocol was in the plasmid taken for transformation – it contained the gene for LbCas12a expression instead of CcCas9 gene.
“LbCas12a fused with NLS extraction and purification was fully analogous to the protocol for CcCas9 described in Fedorova et al. [14] except the insertion of LbCas12a cDNA instead of CcCas9 cDNA into the pET21a plasmid.” Lines 75-76
The Authors should correct the sentence (lines 59-60) – “In vitro fertilization was performed according to the protocol described in the Zebrafish Book with modifications for sperm media from the cryopreservation protocol” What kind of modifications the Authors did?
We have specified that the difference from the protocol described in the Zebrafish Book was in the sperm media, which recipe was taken from another reference.
“In vitro fertilization was performed according to the protocol described in the Zebrafish Book [15] with modification of zebrafish sperm media from the cryopreservation protocol, described in Matthews et al. [16].” lines 87-88
Please provide the “N” number of animals / group sizes. There is no information about number of replicates.
We have specified the number of analyzed embryos in the S1 Table together with the gRNA sequences (line 241).
There is at least another paper on Cas12a editing in Zebrafish. The authors should discuss the other papers in the field with their results (Liu P, Luk K, Shin M, et al. Enhanced Cas12a editing in mammalian cells and zebrafish. Nucleic Acids Res. 2019;47(8):4169‐ doi:10.1093/nar/gkz184).
The Authors thank the Reviewer for this valuable and relevant point. We have included this paper in the Discussion section.
“TIDE was previously used to estimate both efficiencies [26, 27] and features [28, 29] of editing outcome.” Lines 202-203
Introduction and Discussion section are to short.
We have expanded the manuscripts Introduction and Discussion sections, thoroughly following the comments of the Reviewers.
The introduction section should end with the aim of the paper.
We have underlined the aim of the paper more clearly in the Introduction.
“This article aims in quantitative confirmation of this supposition, requiring comparison of the properties of SpCas9 and LbCas12a in the context of generation of knockout zebrafish for behavioral research.” Lines 63-65
Round 2
Reviewer 1 Report
no comments
Reviewer 2 Report
Authors have kindly addressed all previous comments and suggestions. The manuscript is in sufficient shape for publication.